# Elucidating Gender-Specific Distribution of Imipramine, Chloroquine, and Their Metabolites in Mice Kidney Tissues through AP-MALDI-MSI

**DOI:** 10.3390/ijms25094840

**Published:** 2024-04-29

**Authors:** Md. Monirul Islam, Md Foyzur Rahman, Ariful Islam, Mst. Sayela Afroz, Md. Al Mamun, Md. Muedur Rahman, Md Maniruzzaman, Lili Xu, Takumi Sakamoto, Yutaka Takahashi, Tomohito Sato, Tomoaki Kahyo, Mitsutoshi Setou

**Affiliations:** 1Department of Cellular and Molecular Anatomy, Hamamatsu University School of Medicine, 1-20-1 Handayama, Chuo-Ku, Hamamatsu City 431-3192, Shizuoka, Japan; d23022@hama-med.ac.jp (M.M.I.); d23106@hama-med.ac.jp (M.F.R.); ariful222222@gmail.com (A.I.); e23002@hama-med.ac.jp (M.S.A.); amamun5245@gmail.com (M.A.M.); t.sakamoto0731@gmail.com (T.S.); yutaka.ironman@gmail.com (Y.T.); tsato@hama-med.ac.jp (T.S.); kahyo@hama-med.ac.jp (T.K.); 2Institute of Food and Radiation Biology, Atomic Energy Research Establishment, Bangladesh Atomic Energy Commission, Dhaka 1349, Bangladesh; 3Department of Biochemistry and Microbiology, School of Health and Life Sciences, North South University, Bashundhara, Dhaka 1229, Bangladesh; 4Preppers Co., Ltd., Hamamatsu City 431-3192, Shizuoka, Japan; 5International Mass Imaging Center, Hamamatsu University School of Medicine, 1-20-1 Handayama, Chuo-Ku, Hamamatsu City 431-3192, Shizuoka, Japan; 6Department of Systems Molecular Anatomy, Institute for Medical Photonics Research, Preeminent Medical Photonics Education and Research Center, 1-20-1 Handayama, Chuo-Ku, Hamamatsu City 431-3192, Shizuoka, Japan

**Keywords:** drug, imipramine, chloroquine, AP-MALDI-MSI, kidney

## Abstract

Knowledge of gender-specific drug distributions in different organs are of great importance for personalized medicine and reducing toxicity. However, such drug distributions have not been well studied. In this study, we investigated potential differences in the distribution of imipramine and chloroquine, as well as their metabolites, between male and female kidneys. Kidneys were collected from mice treated with imipramine or chloroquine and then subjected to atmospheric pressure matrix-assisted laser desorption ionization-mass spectrometry imaging (AP-MALDI-MSI). We observed differential distributions of the drugs and their metabolites between male and female kidneys. Imipramine showed prominent distributions in the cortex and medulla in male and female kidneys, respectively. Desipramine, one of the metabolites of imipramine, showed significantly higher (*** *p* < 0.001) distributions in the medulla of the male kidney compared to that of the female kidney. Chloroquine and its metabolites were accumulated in the pelvis of both male and female kidneys. Interestingly, they showed a characteristic distribution in the medulla of the female kidney, while almost no distributions were observed in the same areas of the male kidney. For the first time, our study revealed that the distributions of imipramine, chloroquine, and their metabolites were different in male and female kidneys.

## 1. Introduction

Drug biodistribution of host tissue in animal models helps to validate its compounding location in terms of pharmacokinetics, toxicity, and efficacy of treatments or other substances, especially concerning potential sites of action. To accomplish a desired therapeutic response, a medicine has to first spread across the intended tissue, attach to the proteins on the surface or within the cells, and then trigger a series of interactions of disease process pathways [1]. The pharmacokinetics of drugs in males and females depends on multiple factors, including molecular elements like drug transporters and metabolizing enzymes [2], alongside physiological factors such as lower body weight, larger body fat proportion, diminished glomerular filtration rate, and altered gastric motility [3]. These differences notably influence the increased bioavailability of orally administered drugs, particularly those metabolized by cytochrome P450 (CYP) in females. Additionally, women encounter adverse drug reactions (ADRs) nearly twice as frequently as men [4,5], with elevated blood concentrations and prolonged elimination times observed in females, correlating strongly with sex-specific differences in ADR occurrence. However, men generally exhibit faster renal functions, including glomerular filtration rate, tubular secretion and reabsorption, and hepatic elimination of metabolized drugs [6].

Quantitative whole-body autoradiography (QWBA), a conventional drug distribution determination technique based on radio-labeling (^14^C or ^3^H) [7], can provide high-resolution images of the spatial distribution of drugs on specific tissues with a significant limitation of being unable to differentiate between parent drugs and their metabolites [8]. Bioluminescence uses luciferase protein, whereas fluorescence requires light excitation of fluorophores to produce light, which can then be used for the imaging of drugs [9]. However, both techniques have their limitations, such as autofluorescence, low spatial resolution [10,11], bioluminescence requires a substrate (luciferin), and fluorescence can cause photobleaching and phototoxicity [12]. Another sophisticated method for studying drug distribution and development is liquid chromatography-tandem mass spectrometry (LC-MS/MS), which can detect parent drugs and their potential metabolites using biological matrixes without labeling [13]. This technique cannot provide spatial drug distribution information in tissues or organs [14].

Mass spectrometry imaging (MSI) is a rapidly growing technique used in various biological and clinical applications. It can simultaneously measure multiple analytes while preserving valuable spatial information at the nanometer to micrometer scale [15,16,17,18]. MSI has gained significant interest in studies on drug distribution in tissues [19], primarily due to its advantages over other distribution methods such as bioluminescence and fluorescence, QWBA, and LC-MS/MS. MSI encompasses mainly three techniques, which include desorption electrospray ionization-MSI (DESI-MSI), secondary ionization mass spectrometry (SIMS), and matrix-assisted laser desorption–ionization-MSI (MALDI-MSI). Among the three MSI tools, this study opted for AP-MALDI-MSI (iMScope^TM^ QT) because it offers more robust quantitative capabilities and advanced imaging competencies with its high spatial resolution (down to micrometer scale) and wide range of analyzing capabilities [20]. AP-MALDI-MSI stands out for its speed (up to 32 pixels/s), label-free nature, sensitivity, and spatial resolution (25 μm × 25 μm) [21]. Integrated with the LCMS-9030, iMScope^TM^ QT simplifies sample preparation, providing comprehensive distribution information and quantitative data [21,22]. Therefore, this study was designed to explore the gender-specific distribution of two well-known drugs, imipramine and chloroquine, in mice, especially kidneys. Studying gender-specific drug distribution and its metabolic phenomenon is crucial to designing personalized medicine and reducing toxicity.

Imipramine is a tricyclic antidepressant that modulates serotonin and norepinephrine reuptake inhibitors, serving diverse therapeutic purposes such as treating nocturnal enuresis, chronic neuropathic pain, and panic disorder [23,24,25]. Desipramine, the active metabolite of imipramine, is produced by cytochrome P450 enzyme metabolism in the liver (Figure 1A) [26]. The pharmacokinetics of imipramine varies depending on gender. Studies show that women tend to absorb and distribute antidepressants such as imipramine more effectively than men. Women typically have gastrointestinal systems with a higher pH level than men and a significant amount of body fat [27,28,29].

Chloroquine, renowned for its antimalarial and anti-autoimmune effects, impedes heme polymerization into hemozoin and inhibits autophagy, exhibiting potential against diseases like cancer and COVID-19 [30,31,32]. Cytochrome P450 enzymes catalyze the formation of desethylchloroquine, the primary metabolite of chloroquine (Figure 1B) [33]. Chloroquine absorption varies between males and females, potentially influenced by differences in gastrointestinal functions or hormonal levels. However, other pharmacokinetic aspects like half-life, maximum concentration, area under the curve (AUC), clearance, and distribution of chloroquine appear to be comparable between the two genders [34].

Understanding gender-specific pharmacokinetics in the kidneys is important for designing personalized medicine and reducing toxicity. To our knowledge, gender-specific distributions of imipramine and chloroquine and their metabolites in kidneys have not yet been reported. Therefore, we aimed to elucidate the gender-specific distributions of imipramine and chloroquine and their metabolites in mice kidneys by applying AP-MALDI-MSI.

## 2. Results

### 2.1. Detection of Standard Imipramine and Chloroquine Drugs Applying AP-MALDI-MSI

In this study, we chose imipramine and chloroquine to investigate the potential differences in drug distribution between male and female kidneys. We first optimized the MSI conditions for the visualization of these drugs using pure compounds. As expected, both standard drugs were detected as protonated ions in the mass spectra. Imipramine was detected at *m*/*z* 281.20 [M+H]^+^ (Figure 2A), while chloroquine was detected at *m*/*z* 320.18 [M+H]^+^ (Figure 2B). The ion images were clearly visualized at concentrations up to 0.01 μg/mL (0.3 μL/spot on glass slide) (Figure 2C). 

### 2.2. Detection of Imipramine and Chloroquine in Treated Mice Kidneys Using AP-MALDI-MSI

Next, we performed MSI with the control group (saline-treated) and imipramine-treated male and female mice kidney sections. Imipramine was detected at *m*/*z* 281.20 [M+H]^+^ in the average mass spectra acquired from imipramine-treated kidney sections. In addition, two metabolites of imipramine, namely desipramine and 2-hydroxy imipramine, were detected at *m*/*z* 267.18 [M+H]^+^ and *m*/*z* 297.19 [M+H]^+^, respectively (Figure 3C,D). No background or interfering peak was observed at these *m*/*z* values in the mass spectra of control kidney sections (Figure 3A,B).

Finally, we analyzed the kidneys of control and chloroquine-treated male and female mice with MSI. Chloroquine and two its metabolites, desethylchloroquine and chloroquine M (-N(C_2_H_5_)_2_), were detected at *m*/*z* 320.18 [M+H]^+^, *m*/*z* 292.15 [M+H]^+^, and *m*/*z* 247.12 [M+H]^+^, in the mass spectra of chloroquine-treated male and female kidney sections, respectively (Figure 3E,F). No peaks at these *m*/*z* values were detected in the mass spectra of control kidney sections (Figure 3A,B).

### 2.3. Spatial Distribution of Imipramine and Its Metabolites in the Kidneys of Treated Mice

Ion images of imipramine and its metabolites showed a clear difference in their distributions between male and female kidneys (Figure 4A,C). In the control group of male and female kidneys, no ion images of the parent drug and its metabolites were observed (Figure 4B,D). As seen in the box plots, imipramine was prominently distributed in the cortex of the male kidney (Figure 5A). On the other hand, the female kidney showed an accumulation of imipramine in the medulla, but its distribution was significantly higher (*** *p* < 0.001) in the male kidney compared to those in the female kidney (Figure 5B). Desipramine showed characteristic distributions in the medulla in both male and female kidneys, but the distribution was significantly higher (*** *p* < 0.001) in the male kidney compared to that of the female kidney (Figure 5B).

### 2.4. Spatial Distribution of Chloroquine and Its Metabolites in the Kidneys of Treated Mice

We have found that chloroquine and its metabolites have distinct localization patterns in the renal structure of male and female mice. Chloroquine and its metabolites showed a characteristic accumulation in the pelvis areas of both male and female kidneys; interestingly, their distributions in the medulla clearly distinguished between male and female kidneys (Figure 6A,C), respectively. Prominent distribution of the parent drug and its metabolites was observed in the medulla of the female kidney (Figure 6C). On the other hand, almost no distributions were observed in the medulla of the male kidney (Figure 6A). None of the ion images of chloroquine and its metabolites in the control group were detected (Figure 6B,D). As seen in the box plots, the distributions of chloroquine and its metabolites were significantly higher (*** *p* < 0.001) in the medulla of female kidneys compared to the male kidneys (Figure 7B). We also observed that the characteristic distribution of chloroquine and its metabolites were detected in the pelvis of both male and female kidneys, respectively, but the distributions were significantly higher in the female pelvis (*** *p* < 0.001) than in the male (Figure 7C).

## 3. Discussion

Gender plays a key factor in the risk of adverse responses to drugs. Females are more likely to report experiencing adverse reactions than males. There is an increasing agreement that gender-specific differences in drug pharmacokinetics are a major factor contributing to higher drug toxicity in females. These variations arise from physiological differences such as body composition variations, plasma protein concentrations, and liver and kidney function [35]. Kidneys are frequently involved in the detoxification process by being exposed to drugs and their metabolites in the bloodstream; drug-induced nephrotoxicity in the kidney is a common, widespread clinical concern [36,37]. The rising occurrence of kidney disease, which commonly affects the processing of various drugs in the body, highlights the importance of drugs as an essential form of treatment for individuals with kidney disease [38]. It is essential to monitor kidney function, especially when administering drugs known to affect renal tissue, to prevent and manage potential kidney damage.

Two common metabolic and excretory phenomena may occur during drug biotransformation. The first involves drug metabolism in the liver; then, the metabolites accumulate in the kidney. The second process involves both drug metabolism and accumulation in the kidney. In the kidneys, drugs are metabolized through a process often involving enzymes, notably cytochrome P450. Currently, 57 potentially functional full-length CYP genes in humans and 102 genes in mice have been identified [39]. Among these enzymes, CYP1, CYP2, CYP3, and CYP4 families predominantly metabolized drugs and various non-drug xenobiotics [40], while enzymes from the other CYP P450 families are crucial in synthesizing and metabolizing endogenous compounds. Human CYP2D6 genes homologous to mice genes CYP2D22 are responsible for desipramine metabolism, one of the metabolites of imipramine [41]. Studies have shown that CYP2D6 activity differs from male to female, with higher activity observed in females compared to males [42,43]. Our study found that there was a distinct difference in the distribution of imipramine and its metabolites between male and female kidneys. Imipramine and its metabolites were observed in the cortex of the male kidney. At the same time, desipramine and 2-hydroxy imipramine were accumulated in the renal medulla of male and female kidneys (Figure 4A,C). In the female kidney, imipramine, desipramine, and 2-hydroxy imipramine were also accumulated in the renal medulla, but their distributions were higher in male than female kidneys (Figure 5B). Previous studies have shown that the accumulation of imipramine and its metabolites in the kidney could cause potential toxic effects, such as glomerulonephritis and inflammatory cell infiltration. Moreover, these components may also diminish antioxidant levels, leading to oxidative damage in the kidney [44]. Imipramine, along with desipramine and 2-hydroxy imipramine, may induce the release of inflammatory cytokines interleukin-1β (IL-1β) and tumor necrosis factor-alpha (TNF-α) and nitric oxide from microglia [45].

Human CYP2C8 genes are homologous to mice genes CYP2C65, responsible for chloroquine metabolism [40,41]. Previous studies have identified that the CYP isoforms (CYP2C8 and CYP3A3/5) were involved in the metabolism of chloroquine to desethylchloroquine using human liver microsomes [46]. Studies indicated that 52% of the CYP species among male mice were most highly expressed in the liver, while 10% were found in the kidney. Additionally, differences in expression between genders were observed in 29 CYPs, with 24 showing higher expression in females than males [40]. Our experiment found that the distribution of chloroquine, desethylchloroquine, and chloroquine-M (-N(C_2_H_5_)_2_) varied in the kidneys of males and females. Chloroquine and its metabolites were predominantly accumulated in the pelvis of male and female kidneys (Figure 6A,C). Interestingly, they showed a distinct distribution in the medulla of the female kidney (Figure 6C), while almost no distributions were found in the corresponding regions of the male kidney (Figure 6A). Previous findings indicated a potential association between chloroquine and its metabolites with renal damage, as they can cause oxidative damage by stimulating lipid peroxidation and inhibiting the activity of antioxidant enzymes [47,48]. 

There are several possibilities explaining these gender-specific differences in the accumulation of drugs in the kidneys. One could be fundamental differences in the mechanism of sex hormones, such as estrogen and testosterone, which can influence renal function and expression of drug transporters in the kidneys. It has been suggested that female sex hormones could affect drug treatment; for example, estrogens might boost the response to selective serotonin reuptake inhibitors or hinder the response to tricyclic antidepressants [49]. Studies also showed that the association between serum testosterone and kidney function differs between genders, with higher testosterone levels associated with better kidney function in men but worse kidney function in women [50]. Other reasons included differences in the expression or activity of renal transporters, which are responsible for moving drugs in and out of cells and can lead to sex-specific variations in kidney drug distribution [51]. Another factor is the glomerular filtration rate (GFR), which indicates how well the kidneys filter blood and drugs, which is directly related to body weight, and typically, women are smaller than men, thus increasing clearance in men [52,53].

This research revealed the differences in the distribution of imipramine and chloroquine in the kidneys of mice. These differences may be due to the variances in drug absorption, metabolism, and excretion between male and female mice. In this current study, we just focused on drug distribution in the kidneys of males and females. However, further studies are needed to explore the molecular pathways and physiological factors contributing to the gender-specific differences in drug distribution in specific kidney areas noted in our study. Clinical trials are crucial for examining gender-specific differences in drug distribution in human kidneys. Conducting comparative pharmacokinetics studies in humans can determine if there are similar differences in drug metabolism and excretion as seen in mice. These findings could lead to the development of personalized medicine for male and female patients.

## 4. Materials and Methods

### 4.1. Chemicals and Materials

Standards of imipramine hydrochloride (Lot no: SKL0836, molecular weight: 316.87 g/mole) and chloroquine diphosphate (Lot no: LEE6907, molecular weight: 515.86 g/mole) were purchased from Fujifilm Wako Pure Chemical Industries (Osaka, Japan). Hematoxylin, eosin, xylene, pathomount, LC-MS grade ultrapure water, and ethanol were also purchased from Fujifilm Wako Pure Chemical Industries (Osaka, Japan). Super cryo-embedding medium (SCEM) and optimum cutting temperature (OCT) compounds were purchased from Waters, Section Lab, and Sakura Finetek Japan, respectively. Matrix α-cyano-4-hydroxycinnamic acid (CHCA) was purchased from Sigma-Aldrich (St. Louis, MO, USA).

### 4.2. Preparation of Standard Solution for AP-MALDI-MSI

Stock of imipramine and chloroquine (1 mg/mL) were prepared in ethanol-water (80:20, *v*/*v*) and acetonitrile–water (80:20, *v*/*v*), respectively. The stock solutions were diluted with methanol–water (50:50, *v*/*v*) to obtain working solutions of 0.01, 0.1, and 1.0 μg/mL, respectively. These working solutions were spotted on an indium tin oxide (ITO)-coated (100 Ω, Matsunami, Osaka, Japan) glass slide (0.3 μL/spot) separately and allowed to dry at room temperature. After that, an equal volume of CHCA (10 mg/mL) in acetonitrile-water (50:50, *v*/*v*) was applied to the standard spots. After drying, the AP-MALDI-MSI experiments were carried out using the parameters mentioned in Section 4.5.

### 4.3. Animals and Experimental Design

Wild-type male and female C57BL/6J mice aged 19 weeks (about 25–30 g) were used for this experiment. The mice were bred in a standard animal facility with climate-controlled environmental conditions (temperature: 22 ± 2 °C, light/dark cycle: 12 h with the duration of light from 7 a.m. to 7 p.m.). Mice were divided into the following groups: (i) control (normal saline-treated; male, *n* = 2, and female, *n* = 2), (ii) imipramine-treated male mice (*n* = 2), imipramine-treated female mice (*n* = 2), (iv) chloroquine-treated male mice (*n* = 2), and (iv) chloroquine-treated female mice (*n* = 2). Each drug (dissolved in normal saline water) was injected intraperitoneally (IP) at a dose of 30 mg/kg of body weight. Mice were sacrificed by cervical dislocation after two hours of drug administration. Kidney samples were collected immediately, embedded in SCEM, and then rapidly frozen in dry ice. The frozen kidney samples were subsequently stored at −80 °C until sectioning. This study was conducted according to the Institutional Animal Care and Use Committee of Hamamatsu University School of Medicine guidelines, and the institutional review board approved all experimental procedures using living animals. Every possible effort was undertaken to reduce the number of animals used and any potential distress.

### 4.4. Sample Preparation for AP-MALDI-MSI Measurements

Cryosection of mice kidneys was done using the cryostat (CM1950; Leica, Wetzlar, Germany). Before being sectioned, the frozen kidney samples (control, imipramine, and chloroquine treated) were kept inside the cryostat chamber at −20 °C for 30 min. Then, kidney samples were mounted on a sample holder using the optimum cutting temperature (OCT) compound. After trimming at the optimum level, expected sagittal sections of kidneys were collected to a thickness of 10 μm. Tissue sections were mounted onto precooled Indium Tin Oxide (ITO) coated glass slides (100 Ω, Matsunami, Osaka, Japan) and fixed by gently pressing a finger straight opposite onto the section. Then, the slides were dried before applying the matrix using a desiccator. A layer (0.7 μm thickness) of CHCA matrix was deposited onto tissue sections by evaporating the matrix powder at 250 °C under vacuum conditions using iMLayer^TM^ (Shimadzu, Kyoto, Japan).

### 4.5. AP-MALDI-MSI

MSI was conducted using an imaging mass microscope (iMScope^TM^ QT Shimadzu, Japan) equipped with an AP-MALDI source, a quadrupole time-of-flight mass spectrometer, and an integrated optical microscope. Before using MSI, calibration was performed externally using sodium iodide (400 μg/mL in 50:50 methanol-water (*v*/*v*)) at the mass range of *m*/*z* 100−2800, and the mass resolving power was 19,896 (at *m*/*z* 172.88). The AP-MALDI-MSI experiments were carried out in positive ion mode over the range of *m*/*z* 150–550. The following parameters were optimized as follows: pitch size of 25 μm, a laser diameter of around 25 μm, a laser intensity of 70%, a laser shot number of 100, a detector voltage of 2.20 kV, a repetition rate of 1000 Hz, a heat block temperature of 450 °C, and a DL temperature of 250 °C. The data was acquired by Imaging MS Solution software (version 2.00.00A, Shimadzu, Japan).

### 4.6. Hematoxylin and Eosin Staining

After AP-MALDI-MSI measurement, the same sections of sagittal kidney tissues were stained with hematoxylin and eosin (H&E) dyes to facilitate anatomical localization. Before staining, the post-irradiated tissue sections were washed with acetone for a few seconds to remove the matrix. Then, the tissue samples were placed into hematoxylin for 5 min. Afterward, the slides were rinsed with tap water for 1 min, and water was applied to the opposite side of the glass slide rather than directly on the tissue. Follow with a 1-min rinse in 80% ethanol (80:20, ethanol-water (*v*/*v*)). After that, it was submerged in an eosin solution for 30 s to stain the tissue sections. The slides were then washed in distilled water for 1 min. Dehydrated the tissue by passing the slide through a series of increasing ethanol concentrations (80%, 90%, 100%, and 100% ethanol) for 1 min in each, respectively. Finally, clear the slide with a clearing agent (e.g., xylene) for 3 min to remove any remaining water. After allowing sufficient time for air drying, a mounting medium (e.g., pathomount) was applied to the slide to protect tissue from oxidation and to attach tissue with a cover slip. A coverslip was applied over the mounting medium to avoid air bubble formation around the tissue. Then, the stained slide image was acquired by a digital slide scanner, the Hamamatsu NanoZoomer S60 (Hamamatsu Photonics, Hamamatsu, Japan).

### 4.7. MSI Data Analysis

IMAGEREVEAL^TM^ MS (version 1.20.0.10960, Shimadzu, Japan) software was used to analyze and process the data. Data was normalized by total current ion (TIC). Other parameters for data processing were as follows: tolerance/bin size of 0.01 Da and threshold value of 0%. The H&E image of the same section was registered and superimposed on the ion images to create a region of interest (ROIs). The intensity values of each pixel from each ROI were exported to MS Excel (Office 2021) for box-plot preparation and further analysis. Mann–Whitney U-test was used for statistical data analysis because the data is not equally distributed between the male and female groups. The H&E image was processed using NDP.view 2 (version U12388-01, Hamamatsu Photonics, Hamamatsu, Japan) software.

## 5. Conclusions

This study was conducted to investigate the gender-specific pharmacokinetics of imipramine and chloroquine as well as their metabolites in mice kidneys by employing a highly sensitive AP-MALDI-MSI technique. Interestingly, we observed different distribution patterns of those drugs and metabolites between male and female kidneys. A notable distribution of imipramine was found in the cortex of kidneys and medulla of female kidneys. On the other hand, chloroquine and its metabolites showed characteristic distributions exclusively in the medulla of the female kidneys. Our findings highlight the importance of considering gender-specific pharmacokinetics during drug development. Incorporating gender-specific analyses in preclinical studies could lead to the development of safer and more effective drugs for both men and women. The study opens avenues for further research into gender-specific pharmacokinetics and its implications for drug therapy.

## Figures and Tables

**Figure 1 ijms-25-04840-f001:**
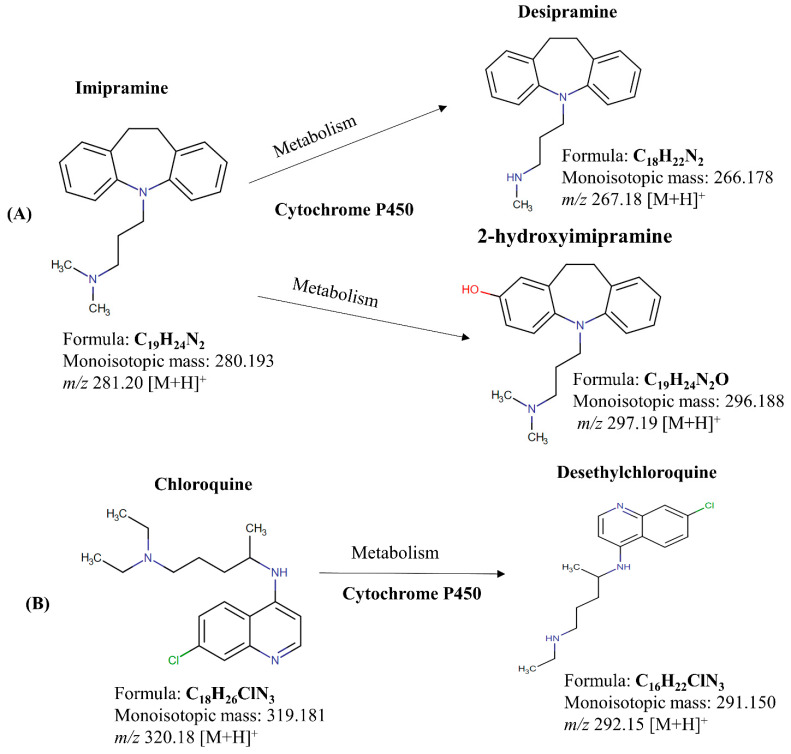
Chemical structure of (**A**) imipramine and its metabolites desipramine and 2-hydroxy imipramine, (**B**) chloroquine and its metabolite desethylchloroquine.

**Figure 2 ijms-25-04840-f002:**
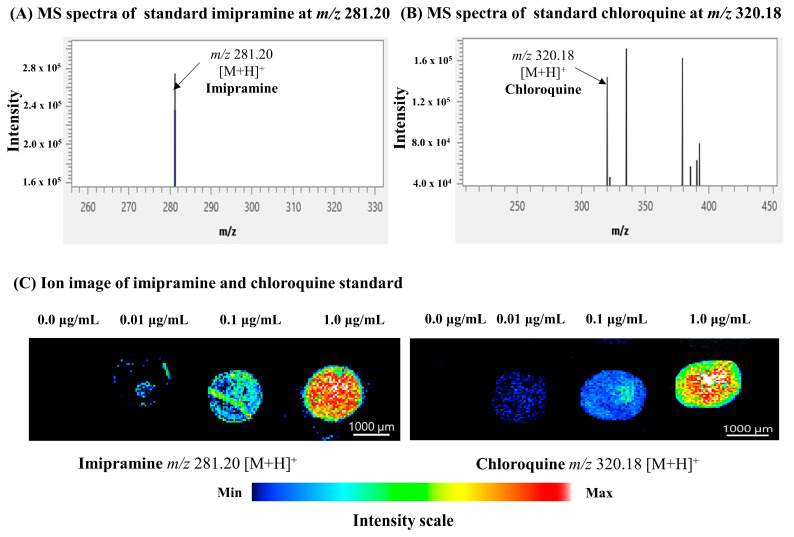
Mass spectra and ion images of standard imipramine and chloroquine. (**A**,**B**) Representative mass spectra were acquired from standard imipramine and chloroquine were spotted on ITO-coated glass slides. (**C**) The ion images of imipramine (at *m/z* 281.20 [M+H]^+^) and chloroquine (at *m*/*z* 281.20 [M+H]^+^) standards were obtained at 25 μm spatial resolution.

**Figure 3 ijms-25-04840-f003:**
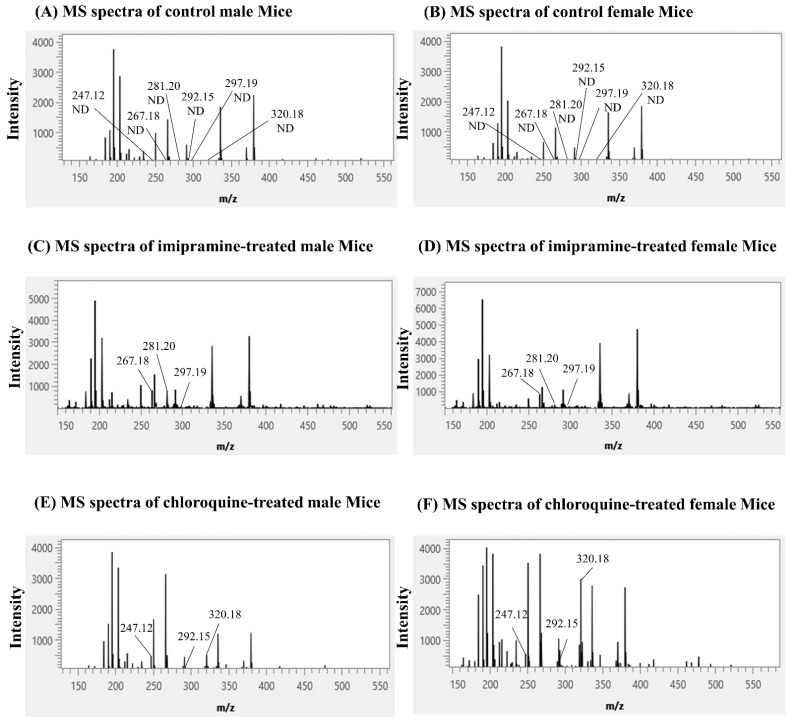
Mass spectra of imipramine and chloroquine and their metabolites. (**A**,**B**) Mass spectra were acquired from control groups. (**C**,**D**) Mass spectra were acquired from imipramine-treated male and female kidneys. (**E**,**F**) Mass spectra were acquired from chloroquine-treated male and female kidneys. ND: not detected.

**Figure 4 ijms-25-04840-f004:**
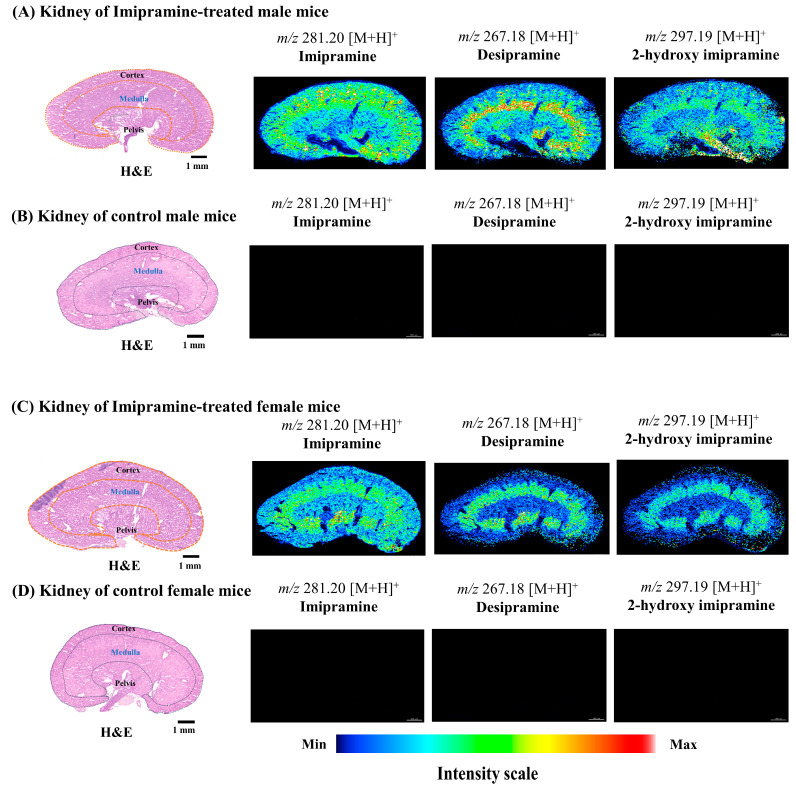
Localization of imipramine, desipramine, and 2-hydroxy imipramine in male and female mice kidneys. (**A**,**C**) ion images of imipramine and its metabolites in imipramine-treated male and female kidneys. (**B**,**D**) ion images of imipramine and its metabolites in control groups (no peaks were observed). Here, spatial resolution was 25 µm × 25 µm (X, Y).

**Figure 5 ijms-25-04840-f005:**
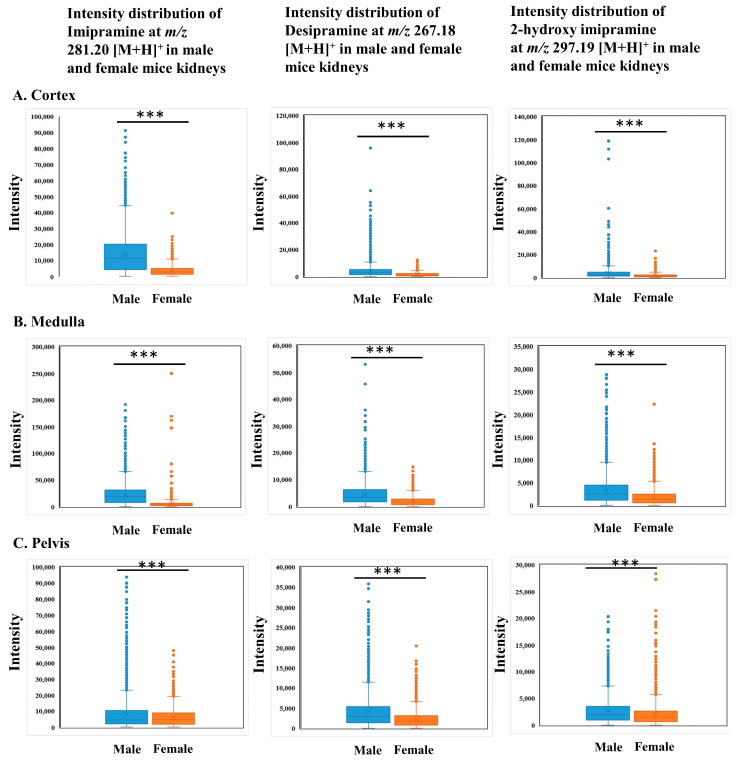
Box plots showing the region-specific distributions of imipramine, desipramine, and 2-hydroxy imipramine in male and female mice kidneys. (**A**) imipramine intensity distribution (*m*/*z* 281.20 [M+H]^+^) in the male and female kidneys (cortex, medulla, and pelvis, top to bottom). (**B**) Desipramine intensity distribution (*m*/*z* 267.18 [M+H]^+^) in the male and female kidneys (cortex, medulla, and pelvis, top to bottom). (**C**) 2-hydroxy imipramine intensity distribution (*m*/*z* 297.19 [M+H]^+^) in the male and female kidneys (cortex, medulla, and pelvis, top to bottom). In the Mann–Whitney U-test, all kidney regions showed significant differences (*** *p* < 0.001) between male and female kidneys.

**Figure 6 ijms-25-04840-f006:**
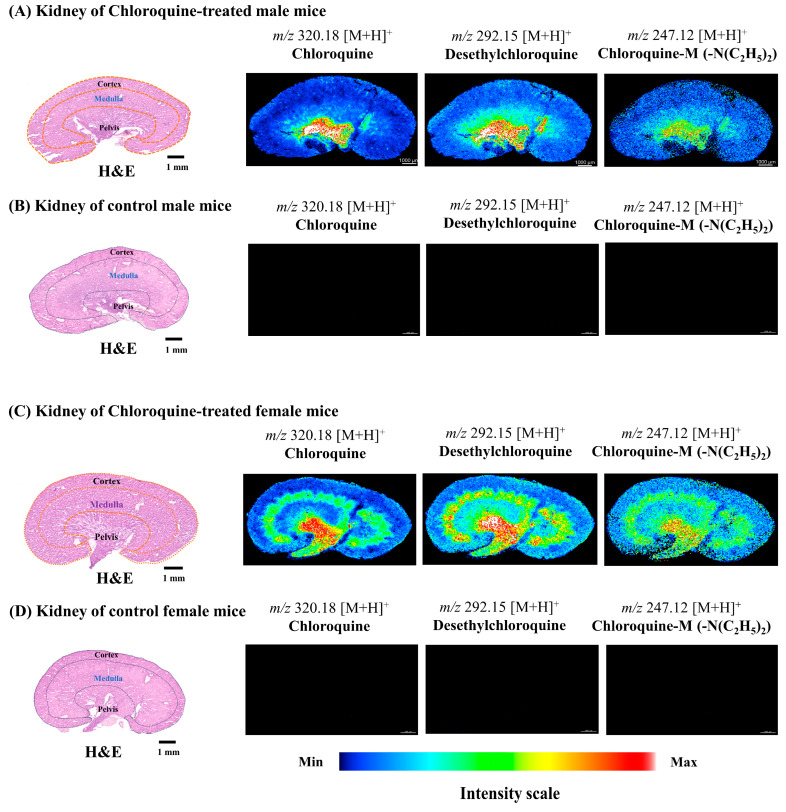
Localization ion images of chloroquine, desethylchloroquine, and chloroquine M(-N(C_2_H_5_)_2_) in male and female mice kidneys. (**A**,**C**) Ion images of chloroquine and its metabolites in chloroquine-treated male and female kidneys. (**B**,**D**) Ion images of chloroquine and its metabolites in control groups (no peaks were observed). Here, spatial resolution was 25 µm × 25 µm (X, Y).

**Figure 7 ijms-25-04840-f007:**
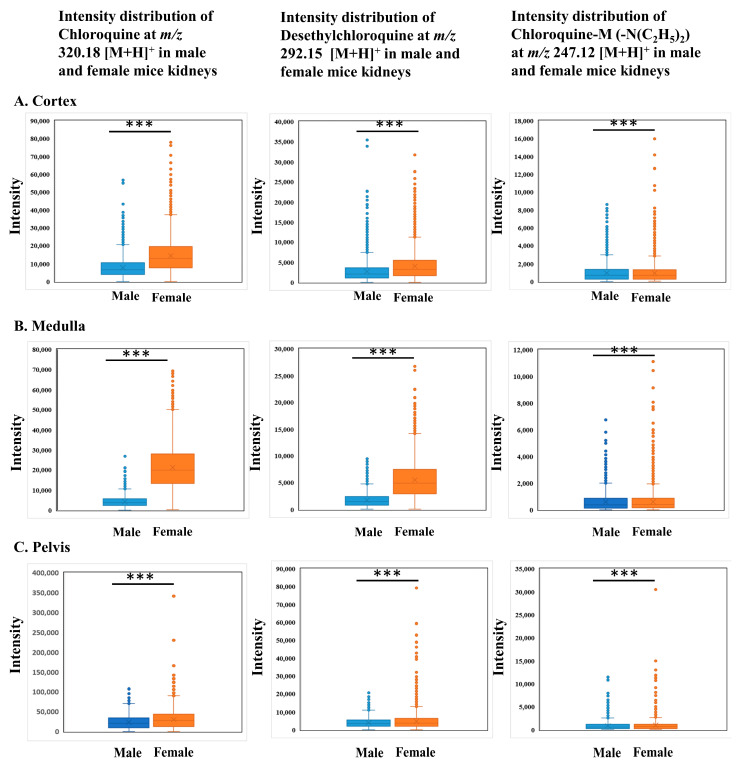
Box plots showing the region-specific distributions of chloroquine, desethylchloroquine, and chloroquine -M (-N(C_2_H_5_)_2_) in male and female mice kidneys. (**A**) Chloroquine intensity distribution (*m*/*z* 320.18 [M+H]^+^) in the male and female kidneys (cortex, medulla, and pelvis, top to bottom). (**B**) Desethylchloroquine intensity distribution (*m*/*z* 292.15 [M+H]^+^) in the male and female kidneys (cortex, medulla, and pelvis, top to bottom). (**C**) Chloroquine-M (-N(C_2_H_5_)_2_) intensity distribution (*m*/*z* 247.12 [M+H]^+^) in the male and female kidneys (cortex, medulla, and pelvis, top to bottom). Mann–Whitney U-test, all regions of kidneys showed a significant difference (*** *p* < 0.001) between male and female kidneys.

## Data Availability

The article contains all the necessary data. Further supporting data will be provided upon a written request addressed to the corresponding author.

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
