# Peer review of "Elucidating Gender-Specific Distribution of Imipramine, Chloroquine, and Their Metabolites in Mice Kidney Tissues through AP-MALDI-MSI"

_ijms, 2024, doi:10.3390/ijms25094840_

Round 1
Reviewer 1 Report
Comments and Suggestions for Authors
The topic of the research is interesting. Below you can find my suggestions on how to improve your manuscript.
1) The Result sections begin with how results are obtained. This should be part of the Materials and Methods section.
2) Line 117: imipramine, chloroquine à imipramine and chloroquine
3) Please discuss the existence of peaks in the control group
4) Discussion of results mainly gives facts we know from the literature and speaks again about the results, but does not discuss them in detail. Please rearrange your manuscript so that each section contains what it should and more properly discuss your results.
5) It is convenient for the Materials and Methods section to be before the Results.
6) Line 311: IP is already introduced in line 296
Author Response
Comments of reviewer 1 and our responses
Reviewer: 1
The topic of the research is interesting. Below you can find my suggestions on how to improve your manuscript.
Response: Thank you very much for your nice comment and evaluation. We have revised the manuscript accordingly.
Comment 1: The Result sections begin with how results are obtained. This should be part of the Materials and Methods section.
Our response to comment 1: Thank you very much for your valuable comment. We have revised it according to your suggestion (lines: 123-128).
Comment 2: Line 117: imipramine, chloroquine à imipramine and chloroquine
Our response to comment 2: Thank you very much for your nice comment. According to your advice, we have revised the sentence to read imipramine and chloroquine (line: 123).
Comment 3: Please discuss the existence of peaks in the control group
Our response to comment 3: Thank you very much for your valuable comment. We have mentioned that the MS peaks corresponded to both drugs, and their metabolites were not detected in the control group (lines: 147-148).
Comment 4: Discussion of results mainly gives facts we know from the literature and speaks again about the results but does not discuss them in detail. Please rearrange your manuscript so that each section contains what it should and more properly discuss your results.
Our response to comment 4: Thank you very much for your valuable suggestions. We have reorganized our manuscript and conducted a more thorough analysis and discussion of our results. The results section explains more details in our revised manuscript (lines: 121-217).
Comment 5: It is convenient for the Materials and Methods section to be before the Results.
Our response to comment 5: Thank you very much for your nice comment. According to the journal guidelines, we have placed the Materials and Methods section after the discussion section (line: 301).
Comment 6: Line 311: IP is already introduced in line 296
Our response to comment 6: Thank you very much for your comment. We have revised our manuscript accordingly (lines: 330-331).

Reviewer 2 Report
Comments and Suggestions for Authors
The article titled "Elucidating Gender-Specific Distribution of Imipramine, Chloroquine, and their Metabolites in the Mice Kidney Tissues through AP-MALDI-MSI" presents a study that investigates the distribution of imipramine, chloroquine, and their metabolites within the kidney tissues of male and female mice. The authors utilized atmospheric pressure matrix-assisted laser desorption ionization-mass spectrometry imaging (AP-MALDI-MSI) to analyze the spatial distribution of these compounds. While the research topic is interesting and the study design is appropriate, there are significant issues and deficiencies in the current manuscript that require major revisions. Detailed comments are provided below.
-
The abstract provides a concise summary of the study; however, it lacks clear information on the main findings of the research. Please revise the abstract to include a brief statement of the key results.
-
The introduction provides a general background on drug distribution in renal tissues and the relevance of gender differences. However, it lacks a clear statement of the research objectives and the significance of the study. Please revise the introduction to clearly state the research objectives and explain why investigating gender-specific drug distribution in the kidneys is important.
-
The section discussing the limitations of bioluminescence and fluorescence imaging techniques could be expanded to include a more comprehensive comparison with AP-MALDI-MSI. Please provide a more detailed explanation of the advantages of AP-MALDI-MSI over other imaging techniques.
-
The methodology section lacks crucial details on the sample preparation, instrumental parameters, and data analysis methods used for AP-MALDI-MSI analysis. Please provide a thorough description of the experimental procedures to ensure reproducibility of the study.
-
The results section provides some information on the distribution patterns of imipramine and chloroquine in male and female mice kidneys. However, it lacks statistical analyses to support the observed differences in distribution. Please perform appropriate statistical tests and include the results in the manuscript.
-
The discussion of the results is limited and does not provide a thorough interpretation or insights into the findings. Please expand the discussion to include a comprehensive analysis of the results and discuss their implications in the context of existing literature.
-
The conclusion section is inadequate and does not effectively summarize the study findings. It lacks a clear statement regarding the significance of the results and their potential impact. Please revise the conclusion to provide a meaningful and comprehensive summary of the research.
-
The references cited in the article are insufficient and lack recent literature. Please ensure that the references are up-to-date and relevant to support the statements made in the manuscript.
-
The article lacks a clear statement regarding the novelty and contribution of the research. It is essential to explicitly state the originality and significance of the study in the introduction and conclusion sections.
-
The article does not discuss the potential limitations of the study or address any potential side effects or adverse reactions of imipramine and chloroquine. Please include a section on limitations and potential implications for future research.
- Questions:
- How does AP-MALDI-MSI overcome the limitations of other imaging techniques?
- What are the statistical analyses performed to validate the observed differences in drug distribution?
- Can the findings of this study be extrapolated to humans? If not, what are the limitations in applying the results to human subjects?
- How do the results of this study contribute to the existing knowledge on drug distribution in renal tissues?
- What are the potential side effects or adverse reactions associated with imipramine and chloroquine that should be considered in the discussion?
- What are the implications of this research for personalized medical treatments or dosage adjustments in clinical practice?
- How do the findings of this study align with or diverge from previous research on gender-specific drug distribution in the kidneys?
- Are there any specific recommendations for future research based on the results of this study?
- How does this study contribute to the broader field of pharmaceutical research and drug development?
- Decision: Major Revision
- The study presented in the manuscript addresses an important research topic and utilizes an appropriate methodology. However, there are significant issues and deficiencies that need to be addressed through major revisions. The authors are requested to carefully consider and address all the comments provided above. Once the revisions are made, the manuscript can be reevaluated for further consideration.
Author Response
Comments of reviewer 2 and our responses
Reviewer: 2
The article titled "Elucidating Gender-Specific Distribution of Imipramine, Chloroquine, and their Metabolites in the Mice Kidney Tissues through AP-MALDI-MSI" presents a study that investigates the distribution of imipramine, chloroquine, and their metabolites within the kidney tissues of male and female mice. The authors utilized atmospheric pressure matrix-assisted laser desorption ionization-mass spectrometry imaging (AP-MALDI-MSI) to analyze the spatial distribution of these compounds. While the research topic is interesting and the study design is appropriate, there are significant issues and deficiencies in the current manuscript that require major revisions.
Response: Thank you very much for your nice comments and evaluations. We have massively revised the manuscript accordingly.
Comment 1: The abstract provides a concise summary of the study; however, it lacks clear information on the main findings of the research. Please revise the abstract to include a brief statement of the key results.
Our response to comment 1: Thank you very much for your precious comment. According to your suggestion, we have revised the abstract and included a brief statement of the key results (lines: 25-39).
Comment 2: The introduction provides a general background on drug distribution in renal tissues and the relevance of gender differences. However, it lacks a clear statement of the research objectives and the significance of the study. Please revise the introduction to clearly state the research objectives and explain why investigating gender-specific drug distribution in the kidneys is important.
Our response to comment 2: Thank you very much for your comment and nice suggestion. We have revised the manuscript massively and mentioned this point in the introduction (lines: 90-93, lines: 110-115).
Comment 3: The section discussing the limitations of bioluminescence and fluorescence imaging techniques could be expanded to include a more comprehensive comparison with AP-MALDI-MSI. Please provide a more detailed explanation of the advantages of AP-MALDI-MSI over other imaging techniques.
Our response to comment 3: Thank you very much for your valuable comment. According to your suggestions, we have mentioned the advantages of AP-MALDI-MSI over other imaging techniques in the introduction (lines: 81-90).
Comment 4: The methodology section lacks crucial details on the sample preparation, instrumental parameters, and data analysis methods used for AP-MALDI-MSI analysis. Please provide a thorough description of the experimental procedures to ensure reproducibility of the study.
Our response to comment 4: Thank you very much for your precious comment.
We have corrected the methodology section according to your suggestion. In detail, we have also mentioned the sample preparation, instrumental parameters, and experimental procedures for AP-MALDI-MSI to ensure the study's reproducibility (lines: 313-362, lines: 382-391).
Comment 5: The results section provides some information on the distribution patterns of imipramine and chloroquine in male and female mice kidneys. However, it lacks statistical analyses to support the observed differences in distribution. Please perform appropriate statistical tests and include the results in the manuscript.
Our response to comment 5: Thank you very much for your valuable suggestion. Based on your comment, we performed a statistical analysis. For this, the H&E image of the same section was registered and superimposed on the ion images to create region of interests (ROIs). The intensity values of each pixel from each ROI were exported to MS Excel (Office 2021) for box-plot preparation and further analysis. Mann-Whitney U-test was used for statistical data analysis due to the data is not equally distributed between male and female group (lines: 383-391, lines: 168-173, lines: 197-202).
Comment 6: The discussion of the results is limited and does not provide a thorough interpretation or insights into the findings. Please expand the discussion to include a comprehensive analysis of the results and discuss their implications in the context of existing literature.
Our response to comment 6: Thank you very much for your nice comment. According to your comments, we have expanded the discussion section to include interpretations of results and their implications in the context of existing literature (lines: 219-300).
Comment 7: The conclusion section is inadequate and does not effectively summarize the study findings. It lacks a clear statement regarding the significance of the results and their potential impact. Please revise the conclusion to provide a meaningful and comprehensive summary of the research.
Our response to comment 7: Thank you very much for your valuable comment. We have revised the conclusion to provide a meaningful and comprehensive summary of the research (lines: 393-404).
Comment 8: The references cited in the article are insufficient and lack recent literature. Please ensure that the references are up-to-date and relevant to support the statements made in the manuscript.
Our response to comment 8: Thank you very much for your valuable comment. According to your comment, we have tried to cite up-to-date and relevant references from recent literature.
Comment 9: The article lacks a clear statement regarding the novelty and contribution of the research. It is essential to explicitly state the originality and significance of the study in the introduction and conclusion sections.
Our response to comment 9: Thank you very much for your comment. We have revised the introduction and conclusion sections according to your comments and explained the originality and significance of the study in the introduction and conclusion sections.
Comment 10: The article does not discuss the potential limitations of the study or address any potential side effects or adverse reactions of imipramine and chloroquine. Please include a section on limitations and potential implications for future research.
Our response to comment 10: Thank you very much for your comment and valuable suggestion. According to your comment we have included limitations and potential implications for future research in the last paragraph of the discussion (lines: 290-300).
Additional Questions:
- How does AP-MALDI-MSI overcome the limitations of other imaging techniques?
Response: Thank you very much for your comment. Mass Spectrometry Imaging (MSI) encompasses mainly three techniques, which include desorption electrospray ionization-MSI (DESI-MSI), secondary ionization mass spectrometry (SIMS), and matrix-assisted laser desorption-ionization-MSI (MALDI-MSI). Among the three MSI tools, this study opted to use atmospheric pressure-MALDI-MSI (iMScopeTM QT) because it offers more robust quantitative capabilities and advanced imaging competencies with its high spatial resolution (down to micrometer scale) and wide range of analyzing capabilities. AP-MALDI-MSI stands out for its speed (up to 32 pixels/s), label-free nature, sensitivity, and spatial resolution (25 μm x 25 μm) (lines: 81-90).
- What are the statistical analyses performed to validate the observed differences in drug distribution?
Response: Thank you very much for your valuable suggestion. According to your comment we performed statistical analysis. For this, the H&E image of the same section was registered and superimposed on the ion images to create region of interests (ROIs). The intensity values of each pixel from each ROIs were exported to the MS Excel (Office 2021) for box-plot preparation and further analysis. Mann-Whitney U-test was used for statistical data analysis due to the data is not equally distributed between male and female groups (lines: 386-390).
- Can the findings of this study be extrapolated to humans? If not, what are the limitations in applying the results to human subjects?
Response: Thank you very much for your important comment. Clinical trials are crucial for examining gender-specific differences in drug distribution in human kidneys. By conducting comparative pharmacokinetic studies in humans, we can determine if there are similar differences in drug metabolism and excretion as seen in mice (lines: 296-299).
- How do the results of this study contribute to the existing knowledge on drug distribution in renal tissues?
Response: Drug and metabolite accumulation in kidneys can have a toxic effect that causes renal injuries, which is well known. There is a difference in the toxicity of these drugs between male and female patients; this has not been studied yet. This study will contribute to improving our understanding of the further consideration of gender-specific drug distribution and their toxicity in kidneys in future drug development.
- What are the potential side effects or adverse reactions associated with imipramine and chloroquine that should be considered in the discussion?
Response: Thank you very much for your valuable suggestion. According to your suggestion, we have included the adverse association of imipramine and chloroquine in the kidneys in the discussion section.
For Imipramine:
Previous studies have shown that the accumulation of imipramine and its metabolites in the kidney could cause potential toxic effects, such as glomerulonephritis and inflammatory cell infiltration. Moreover, these components may also diminish antioxidant levels, leading to oxidative damage in the kidney [Chang et al., 2021]. Imipramine and its metabolites may induce the release of inflammatory cytokines (IL-1β and TNF-α) and nitric oxide from microglia [Obuchowicz et al., 2020]. (lines: 250-256).
For Chloroquine:
Previous findings indicated a potential association between chloroquine and its metabolites with renal damage, as they can cause oxidative damage by stimulating lipid peroxidation and inhibiting the activity of antioxidant enzymes [Murugavel and Pari 2004, Liao et al., 2022]. (lines: 269-272).
- What are the implications of this research for personalized medical treatments or dosage adjustments in clinical practice?
Response: This study can contribute to adjusting the dose of imipramine and chloroquine for males and females and can also help develop personalized medicine in the future.
- How do the findings of this study align with or diverge from previous research on gender-specific drug distribution in the kidneys?
Response: This study found differential accumulation of imipramine, chloroquine, and their metabolites in male and female mice kidneys. This study is consistent with the previous report that demonstrated the acceleration of renal impairment mediated by imipramine and chloroquine [Chang et al., 2021]; and [Murugavel and Pari 2004, Liao et al., 2022].
- Are there any specific recommendations for future research based on the results of this study?
Response: Thank you very much for your valuable comment. This research revealed the differences in the distribution of imipramine and chloroquine in mice kidneys. This difference may be due to variances in drug absorption, metabolism, and excretion between male and female mice. In this current study, we just focused on drug distribution in male and female mice. However, further studies are needed to explore the molecular pathways and physiological factors contributing to the gender-based differences in drug distribution in specific kidney areas noted in our study (lines: 290-296).
- How does this study contribute to the broader field of pharmaceutical research and drug development?
Response: In pharmaceutical research, only males are considered for the optimization of the doses of drugs during drug development. However, the pharmacokinetics of drugs are different in males and females. Our study also makes that concept clear from imipramine and chloroquine. Not only these two drugs, but other drugs also have such differences. Our study can be used as a reference for further research to explore the differences in the pharmacokinetics of other drugs between male and female subjects. This could lead to optimizing proper drug doses for male and female patients for better drug efficiency and lower toxicity.

Reviewer 3 Report
Comments and Suggestions for Authors
The study investigates the distribution of imipramine, chloroquine, and their metabolites in the kidneys of male and female mice using atmospheric pressure matrix-assisted laser desorption ionization-mass spectrometry imaging (AP-MALDI-MSI). The research aims to elucidate potential sexual dimorphisms in drug distribution within renal tissues, which can have implications for dosage adjustment and therapeutic efficacy.
One notable strength of the study is its innovative use of AP-MALDI-MSI to visualize the spatial distribution of drugs and metabolites within kidney tissues. This technique offers enhanced sensitivity and selectivity, allowing for the precise localization of compounds. Additionally, the investigation of gender-specific differences in drug distribution fills a gap in the scientific literature, as such disparities are not widely reported.
Chemical structures of the studied analyte should be introduced in the tehe text.
While the study identifies gender-specific differences in drug accumulation within the kidneys, it does not delve into the underlying mechanisms driving these disparities. Further research exploring the molecular pathways and physiological factors contributing to sexual dimorphisms in renal drug distribution would provide deeper insights. Explain this aspect in the text.
The study highlights the potential implications of gender-specific drug distribution for therapeutic interventions. However, it remains unclear how these findings translate to clinical practice and whether similar sexual dimorphisms exist in human kidneys. Investigating the translational relevance of these observations would enhance their clinical significance. Explain this aspect in the text.
In conclusion, while the study offers valuable insights into sexual dimorphisms in renal drug distribution using AP-MALDI-MSI, there are opportunities for further exploration and refinement. Addressing the highlighted points would strengthen the research's scientific rigor and enhance its contribution to the field of pharmacology and drug development.
Comments on the Quality of English LanguageEnglish is OK
Author Response
Comments of reviewer 3 and our responses
Reviewer: 3
One notable strength of the study is its innovative use of AP-MALDI-MSI to visualize the spatial distribution of drugs and metabolites within kidney tissues. This technique offers enhanced sensitivity and selectivity, allowing for the precise localization of compounds. Additionally, the investigation of gender-specific differences in drug distribution fills a gap in the scientific literature, as such disparities are not widely reported.
Response: Thank you very much for your nice comments and evaluations. We have revised the manuscript accordingly.
Comment 1: The chemical structures of the studied analyte should be introduced in the text.
Our response to comment 1: Thank you for your valuable comment. According to your suggestion, we have introduced the studied analyte's chemical structures in the manuscript's introduction section (lines: 94-98, lines: 102-105, lines: 115-118).
Comment 2: While the study identifies gender-specific differences in drug accumulation within the kidneys, it does not delve into the underlying mechanisms driving these disparities. Further research exploring the molecular pathways and physiological factors contributing to sexual dimorphisms in renal drug distribution would provide deeper insights. Explain this aspect in the text.
Our response to comment 2: Thank you very much for your nice comment. Based on your valuable advice, we have updated the manuscript and highlighted the above points in our discussion section (lines: 48-53, lines: 219-224).
Comment 3: The study highlights the potential implications of gender-specific drug distribution for therapeutic interventions. However, it remains unclear how these findings translate to clinical practice and whether similar sexual dimorphisms exist in human kidneys. Investigating the translational relevance of these observations would enhance their clinical significance. Explain this aspect in the text.
Our response to comment 3: Thank you for your valuable suggestion. Based on your comment, we revised the manuscript. Clinical trials are crucial for examining gender-specific differences in drug distribution in human kidneys. Comparative pharmacokinetic studies in humans can determine if there are similar differences in drug metabolism and excretion as seen in mice (lines: 296-299).
Comment 4: While the study offers valuable insights into sexual dimorphisms in renal drug distribution using AP-MALDI-MSI, there are opportunities for further exploration and refinement. Addressing the highlighted points would strengthen the research's scientific rigor and enhance its contribution to the pharmacology and drug development field.
Our response to comment 4: Thank you for your valuable comment. Based on your important suggestion, we have reorganized the conclusion part of the manuscript. (lines: 393-404).

Round 2
Reviewer 1 Report
Comments and Suggestions for Authors
The authors gave satisfactory answers to my suggestions. I suggest you accept the manuscript.
Reviewer 2 Report
Comments and Suggestions for Authors
Accepted